# Noise Reconstruction and Removal Network: A New Way to Denoise FIB-SEM Images

## Abstract

Recent advances in Focused Ion Beam-Scanning Electron Microscopy (FIB-SEM) allow the imaging and analysis of cellular ultrastructure at nanoscale resolution, but the collection of labels and/or noise-free data sets has several challenges, often immutable. Reasons range from time consuming manual annotations, requiring highly trained specialists, to introducing imaging artifacts from the prolonged scanning during acquisition. We propose a fully unsupervised Noise Reconstruction and Removal Network for denoising scanning electron microscopy images. The architecture, inspired by gated recurrent units, reconstructs and removes the noise by synthesizing the sequential data. At the same time, the fully unsupervised training guides the network in distinguishing true signal from noise and gives comparable/even better results than supervised approaches on 3D electron microscopy data sets. We provide detailed performance analysis using numerical as well as empirical metrics.

## 1 Introduction

Recent advances in Focused Ion Beam-Scanning Electron Microscopy (FIB-SEM) have led to unprecedented biological tissue visualization and analysis, as well as understanding of cellular ultrastructure and cell-to-cell interactions (Xu et al., 2017). High-resolution FIB-SEM data sets often consist of volumes sliced into thousands of 6K×4K images with 4nm resolution per voxel, allowing a 3D reconstruction of a fraction of tissue volume. Depending on the tissue type, sample preparation, acquisition settings, detector used, etc., the images may contain a significant quantity of noise making any further analysis tedious or even impossible (Kubota et al., 2018; Liu et al., 2018).

By definition, image denoising is the process of taking a noisy image $x$ and separating the noise $n$ from the true signal $s$: $x = s + n$. Following the typical assumption for the noise (Foi et al., 2008; Wu et al., 2019) and taking the microscope's characteristics into account, we can assume that the noise is: (i) a zero-mean random noise, that is for any pixel the noise is a discrete random number added to the pixel 'true value'; (ii) each pixel noise is independent, so the noise value at any pixel does not depend on the noise at any other pixel, but it is signal dependent. While the noise is random and independent, the signal is not and this is what, typically, denoising methods rely on.

In recent years, deep learning methods, and particularly the Convolutional Neural Networks (CNNs), have established themselves as powerful analytical tools in machine learning - to name just a few Redmon & Farhadi (2018); Tan et al. (2020); Chen et al. (2017); Ronneberger et al. (2015); Tao et al. (2020); Sun & Chen (2020). In the field of denoising, CNNs have been very useful (Kim et al., 2019; Liu et al., 2019; Yu et al., 2019) especially when the noise characteristics are unknown, making any mathematical modeling difficult. In this paper we apply a CNN technique to FIB-SEM acquired images, taking into account the relevant specifics.

FIB-SEM allows 3D imaging of biological fine structure at nanoscale resolution: a thin slice of the sample is removed with the ion beam and the newly exposed surface is imaged with the electron beam. That results in a sequence of images containing isotropic voxels down to 4nm.

Traditionally, training neural networks for denoising demands pairs of noisy and clean images (ground truth images). Theoretically, obtaining denoised images is possible by averaging multiple (up to hundreds) acquisitions of the same sample. As mentioned, this is not feasible with FIB-SEM. The challenges with obtaining ground truth images, as in many biological use cases, is motivation

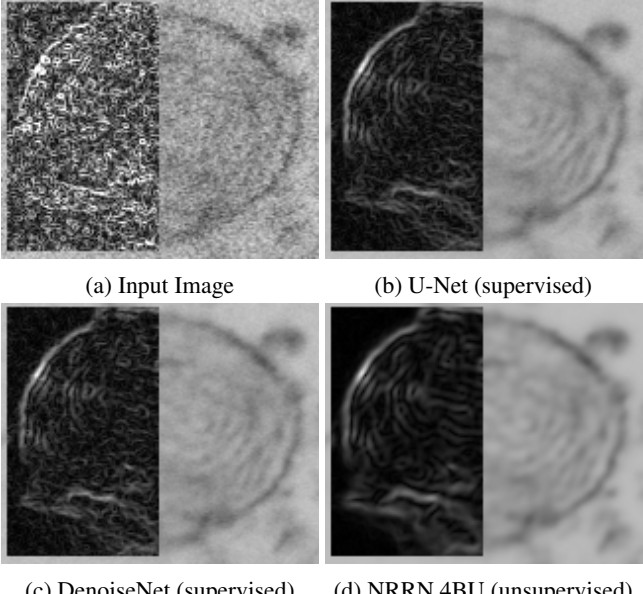

(a) Input Image        (b) U-Net (supervised)

(c) DenoiseNet (supervised)    (d) NRRN 4BU (unsupervised)

Figure 1: The right panel shows the gray original image and left panel shows image after being passed through high-pass filter. A good denoising algorithm should be able to suppress noise while keeping the edges sharp. NRRN's architecture paired with our unsupervised training scheme removes the noise and does a better job at preserving the edges of the organelles as compared to U-Net and DenoiseNet which were trained using supervised techniques.

for developing and utilizing unsupervised techniques, such as a Noise2Noise approach (Wu et al., 2019; Lehtinen et al., 2018). Our network uses a triplet of images as input and is trained to map one noise realization to the other, using our modified Noise2Noise loss function. We refer to the proposed architecture as Noise Reconstruction and Removal Network (NRRN). Fig. 1 shows a visual comparison between NRRN and other state-of-the-art networks trained using supervised techniques. The NRRN is applicable to the case of improving the image quality based on two or three scans of the same slice, or denoising based on the two adjacent slices in the volume stack. Our major three contributions discussed in this paper are:

- a novel noise reconstruction module with soft attention and signal boosting, that upon deployment on large images (more than 24M pixels) homogeneously removes the noise,
- a neural network architecture design using our noise reconstruction module with detailed performance analysis, and
- updated Noise2Noise loss function specifically designed for denoising FIB-SEM data.

## 2   RELATED WORK

Noise-to-clean (N2C) is the traditional supervised learning approach, where the models are trained with pairs of noisy and clean images as inputs and targets respectively. However, when the clean (ground truth) images are not available, the supervised N2C approach is not applicable.

Lehtinen et al. (2018) introduced a new idea: Noise2Noise (N2N). Instead of training a network to map noisy inputs to clean images, their N2N trains on pairs of independently degraded versions of the same training sample.

Wu et al. (2019) progressed the N2N idea with a novel loss function applied to medical images with available pairs of noise realizations. The network is trained to map one noise realization to the other, with a loss function that efficiently combined the outputs from both training subsets. The Noise-to-Void (N2V) training schema (Krull et al., 2019) and the followed Noise-to-self (N2S) (Batson & Royer, 2019) go even a step further and train a network only on a single noisy image. The methods are based on the so-called blind-spot networks where the receptive field of the network does

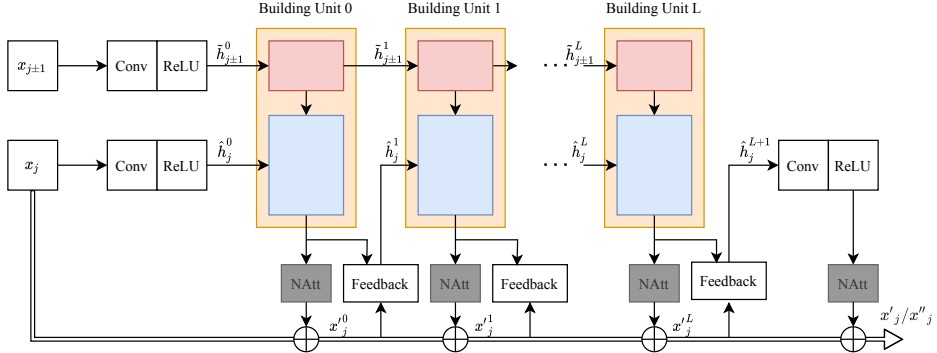

Figure 2: Single Branch Architecture

not include the center pixel. As the blind-spot network has a little bit less information than N2N and N2C, it then expectantly cannot quite reach their performances, but still gives very good results.

Independent of the training scheme, a lot of work has been done on the neural network architectures design. A very popular choice is the U-Net architecture (Ronneberger et al., 2015) and its variations. U-Net was first introduced for segmentation but since has been widely extended in its applications as well as architectures. Alternatively, Remez et al. (2017; 2018) used a CNN that exploits a gradual denoising process. Their DenoiseNet architecture calculates a "noise estimate" that is fine-tuned at each layer based on the previous layer output, and the results are then added to the input image. They have shown that the shallow layers handle local noise statistics, while deeper layers recover edges and enhance textures. Recurrent neural networks (RNN) have been widely developed and used in many applications, particularly with sequential data such as natural language processing and video analysis in order to exploit temporal information. Specifically GRU (Cho et al., 2014; Yao et al., 2015) and ConvLSTM (Shi et al., 2015; Yang et al., 2017) have influenced the NRRN architecture.

## 3 NOISE RECONSTRUCTING AND REMOVAL NETWORK

The proposed Noise Reconstruction and Removal Network (NRRN) architecture and the training framework are designed to take advantage of the sequential nature of FIB-SEM data.

### 3.1 DENOISING MODULE

Typically, a FIB-SEM data set consists of $N$ images representing $N$ consecutive imaged slices $(x_0, ..., x_N)$. Our approach is to obtain a denoised image $\hat{x}_j$ from three adjacent noisy images.[1] To do so, we consider the initial tissue sample into a data set of $N - 2$ independent triplets of images $(x_{j-1}, x_j, x_{j+1})$. We split each triplet into two pairs $(x_{j-1}, x_j)$ and $(x_j, x_{j+1})$ that are passed through two parallel identical branches of the proposed network NRRN.

For a pair of images $(x_j, x_{j\pm1})$, each branch (Fig. 2) consists of:

- one convolution layer ($3 \times 3$ kernel, $64$ channels, and ReLU activation) applied to each of the two images
- $L + 1$ stacked layers made of a building unit (BU) coupled with noise attention (NAtt) and feedback blocks
- a final convolution layer ($3 \times 3$ kernel, $64$ channels, and ReLU activation).

Each BU models noise components (see Fig. 3) by accumulating noise characteristics from previous units and from updated input image after each stage. Within the $BU^l$, the information flows through two major paths - highlighted by an upper and a lower block. The path through the lower block $\hat{h}_j^l$ learns noise features from previous $BU^{l-1}$ and cleaner input $x'^{l-1}_j$. The upper block $\tilde{h}_{j\pm1}^l$, named synthesizer, mostly learns information coming from the adjacent image $x_{j\pm1}$. Motivated by LSTM

---

[1]Or if available, three scans of the same slice

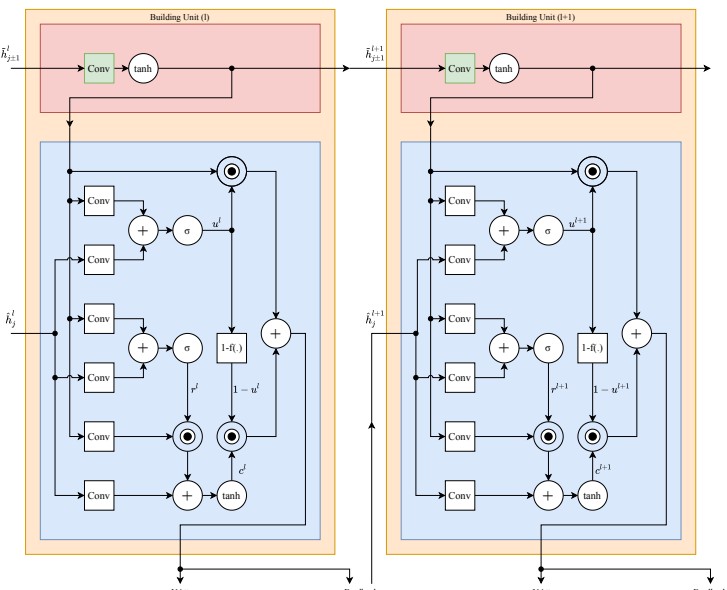

Figure 3: Building Unit

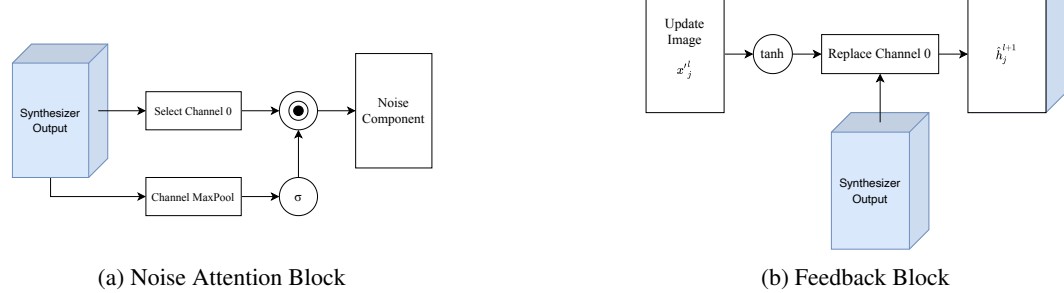

(a) Noise Attention Block

(b) Feedback Block

Figure 4: NRRN Additional Blocks: a) The Noise Attention Block (NAtt), and b) The Feedback Block after the $l^{th}$ BU.

and GRU cells, but with a reduced computational overhead, a $BU^l$ contains two gates: an update gate $u^l$ that decides how much of the previous/adjacent information (coming from $\tilde{h}_{j\pm1}^{l+1}$) and the new one $(1 - u^l)$ to be kept; a reset gate $r^l$ that regulates how much to be forgotten. This allows the synthesizer at every level to accept useful information coming from the $x_{j\pm1}$ while simultaneously being aware about the updated image $x'^l_j$.

Once the synthesizer has learned the noise components, the new features are passed to the noise attention block (NAtt block). The NAtt block filters the BU output and generates noise component which is removed from the corresponding input image to give a new updated image $x'^l_j$ to the next BU. The key equations for every pair of images passed through the BUs are given in the appendix. The NAtt block (see Fig. 4a) receives a feature vector with 64 channels $\hat{h}_j^{l+1}$ as input. It max pools through all channels to create an attention map that gets element-wise multiplied to the first channel of $\hat{h}_j^{l+1}$. We observe that the output of the NAtt block represents component of the noise present in previous noisy image $x'^{l-1}_j$.

Presumably, we obtain a less noisy image $x'^l_j$ after each layer and we want to feed it back to the network. That is achieved in the feedback block (see Fig. 4b) that works as a buffer by gathering information from both the updated image $x'^l_j$ as well as the newly learned features $\hat{h}_j^{l+1}$ from the

previous BU. Thus the first channel of the synthesizer's output $\hat{h}_j^{l+1}$ is updated by the Feedback block giving an input for next BU. For simplicity we use the same notation $\hat{h}_j^{l+1}$, for both: synthesizer output and input of the next BU.

## 3.2 Loss Function

We consider $N - 2$ triplets of images split into two pairs $(x_{j-1}, x_j)$ and $(x_{j+1}, x_j)$. The NRRN network maps each of these pairs to two outputs $x_j' = f(x_{j-1}, x_j, \theta)$ and $x_j'' = f(x_{j+1}, x_j, \theta)$. The two input pairs having independent and zero means noise, generate two denoised versions of the same $j^{th}$ image. The final denoised solution $\hat{x}_j$ of the $j^{th}$ image is an average of the two. One can view, for every slice $x_j$, the adjacent images $x_{j-1}$ and $x_{j+1}$ as discrete versions of the signal along the third direction of the volume. In addition, $x_{j-1}$ and $x_{j+1}$ have their own noise that is spatially independent. Due to the FIB-SEM way of imaging (extremely high isotropic resolution along all three dimensions), we consider that the signal between two slices has good enough properties and we can use a Taylor expansion along the third-axis. The updated loss derived from the Wu et al. (2019) can be rewritten as (for derivation see appendix)

$$L_{n2n} = \frac{1}{N-2} \sum_{j=1}^{N-2} \left\{ \frac{1}{2}||x_j' - x_{j+1}||_2^2 + \frac{1}{2}||x_j'' - x_{j-1}||_2^2 - \frac{1}{4}||x_j' - x_j''||_2^2 \right\} \qquad (1)$$

## 4 Data Sets and Metrics

The NRRN architecture has been specifically designed for FIB-SEM images denoising, and trained and tested on images acquired with a FEI Helios NanoLab 660 DualBeam using an In-Column Detector (ICD) at Oregon Health & Science University (OHSU) Multiscale Microscopy Core (MMC). The OHSU data set is composed of cancer tissue images of dimensions $4K \times 6K$ pixels with $4nm$ resolution. The acquiring technique allowed the creation of "ground truth" images: the tissue sample surface was scanned 10 times before slicing (the number of scans being limited by artifacts produced by extended electron beam scanning), so producing 10 images of the exact same area. The process was repeated 5 times. The "ground truth" images were obtained after applying an affine transformation from an in-house version of the TurboReg algorithm (Thevenaz et al., 1998), removing outliers and finally averaging. Additionally, we experimented with the publicly available data set from EPFL. Here, the ground truth images are not available, and we employed the data set to examine the efficiency of the proposed algorithm in removing artificially added Gaussian and/or Poisson noise.

We use two classical measures in the evaluation process (applicable when ground truth is available): the Peak Signal-to-Noise Ratio (PSNR) and the Structural Similarity Index Measure (SSIM). We also utilize the interquartile range (IQR) of the signal across a straight line in a flat-signal-area. In the FIB-SEM case, the entire workflow from tissue collection to final image acquisition takes roughly two weeks for 1500 images (Riesterer et al., 2020). During this process the clinical specimen undertakes several epoxy of resin infiltration steps to fill the space between the cellular structures. In our analysis we look at the noise presence in the resin because it is a homogeneous material, and as a consequence an "efficient" denoiser should produce a flat signal.

## 5 Results and Comparisons

The OHSU data set was separated into 3 slices for training, 1 for validation and 1 for testing, and the initial large images were cropped into 13650 smaller images of size $256 \times 256$ pixels. The NRRN architecture was implemented in PyTorch (Paszke et al., 2019), with 4 BUs, the image triplets in the training are taken from the same slice, and the training performed with ADAM optimizer for initial learning rate $10^{-4}$, $\beta 1 = 0.9$, $\beta 2 = 0.999$, $\epsilon = 10^{-8}$, and a batch size of 16.

On the test slice, NRRN achieves a PSNR of $31.0197 \pm 0.1905$ dB and a SSIM of $0.9705 \pm 0.0006$. Fig. 5 shows an image from the test slice with a PSNR of 31.11dB, significant reduction of the noise on the resin (the IQR is reduced to $0.94$ - Fig.5 d.), while still keeping sharp edges on the cell organelles with reduced noise (see Fig. 5 e.). Arguably, a model taking triple input is a burden, but inference with double or even single input is possible as discussed in the appendix.

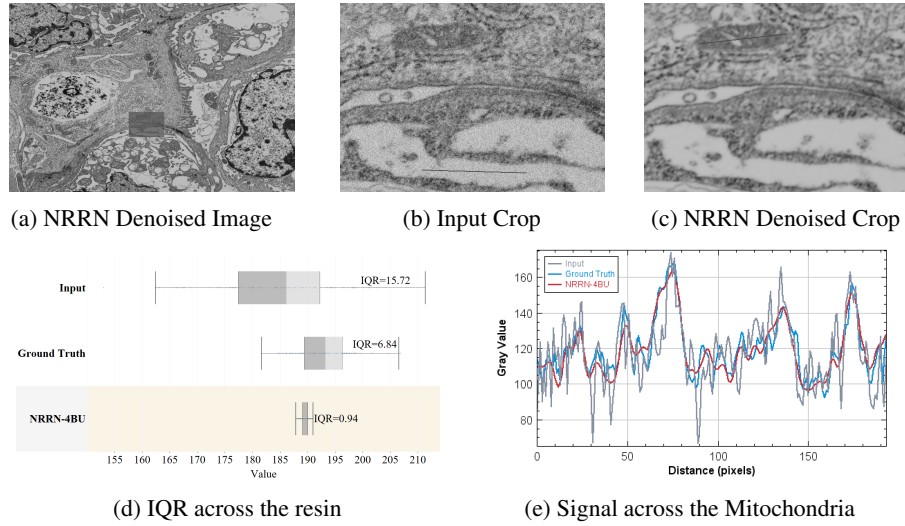

(a) NRRN Denoised Image     (b) Input Crop     (c) NRRN Denoised Crop

(d) IQR across the resin         (e) Signal across the Mitochondria

Figure 5: NRRN denoising results, where the yellow doted line shows where image signal is measured: (a) validation image, (b) zoom in of the noisy input image, (c) zoom in of the denoised image, (d) box plot of the signal across the resin (from subfig. $b$) - showing the significant reduction in the signal/noise variation, and (e) the signal variation across the mitochondria (from subfig. $c$)) - the NRRN denoising keeps sharp edges.

## 5.1 COMPARISONS

We compare NRRN to the following methods:

**Training-free methods:** BM3D, non-local means (NLM), median and Gaussian filters. We applied the NLM to a single image and to an average of 3 (here referred as 'NLM sngl' and 'NLM avg' respectively).

**Training methods:** U-Net and DenoiseNet. We experimented with the classical DenoiseNet (using ground truth images in the training) as presented in Remez et al. (2017) (here referred as 'N2C DenoiseNet- sngl'), but also a version of DenoiseNet that is applicable to a sequence of (three) images Remez et al. (2018) (here referred as 'DenoiseNet' or 'N2C DenoiseNet').

The U-Net is a classical network with a depth of 4 and batch normalization. U-Net is trained on a single image (as input) - 'N2C U-Net sngl' and on an average of three consecutive images 'N2C U-Net avg' (or simply U-Net).

The rather large OHSU FIB-SEM input images ($4K \times 6K$) do not fit into the GPU memory - a typical problem to be faced in this field. A solution is during inference to denoise $256 \times 256$ cropped patches (345 patches per image), and reconstruct the original image. We noticed that even though U-Net overall performs well, the level of denoising depends significantly on the image (patch). On Fig. 6a we plotted the patches PSNR distribution for each of the compared architectures. NRRN removes the noise homogeneously across all patches, with a standard deviation of only $0.52$, only marginally better than DenoiseNet and both outperform significantly the U-Net deviation of $1.26$.

Here, we argue that the loss function ( equation 1) results in better denoising. The ground truth images have better signal than the input images, but they are not noise free. Using per-pixel loss with a ground truth in the training (in the face of L1 or L2 norm) would maximize the PSNR but results in an image that inherits the noise present in the "ground truth". This is avoided by NRRN. On Fig. 6b we compared the noise in the signal flat resin region for U-Net, DenoiseNet and NRRN. The improvements in the IQR are significant for NRRN.

Comparisons on the final whole image based on PSNR and SSIM are shown in table 1. Expectantly, NLM-avg, DenoiseNet and U-Net all outperform NRRN, in terms of PSNR and SSIM. DenoiseNet and U-Net are designed to minimize the MSE loss (equivalent to maximizing the PSNR). In the case

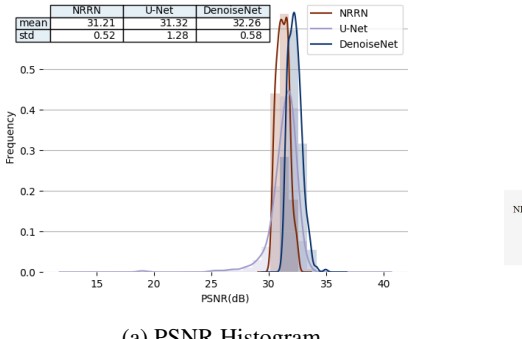

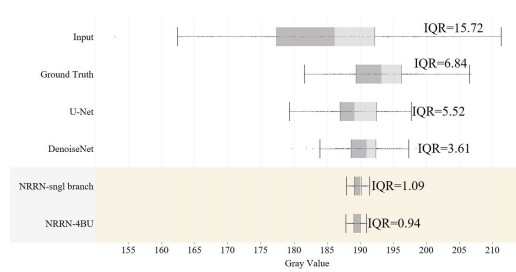

(a) PSNR Histogram            (b) IQR across the resin area

Figure 6: Left Panel: PSNR Histogram for the patches constructing a FIB-SEM image denoised with U-Net, NRRN and DenoiseNet. Right Panel: Comparison of the signal across the resin for U-Net, DenoiseNet, NRRN and the Ground Truth image. NRRN with its specific N2N loss function removes more noise than the N2C trained methods. The IQR of NRRN outperform the rest of the networks.

Table 1: PSNR and SSIM for denoising with training-free methods (left) and training methods (right)

| Training-free | | | Training | | |
|---|---|---|---|---|---|
| **Method** | **PSNR (dB)** | **SSIM** | **Method** | **PSNR (dB)** | **SSIM** |
| Input | 22.9680 | 0.8577 | Input | 22.9680 | 0.8577 |
| NLM -avg. input | **31.3589** | **0.9725** | N2C DenoiseNet | **32.2029** | **0.9779** |
| BM3D | 28.2515 | 0.8747 | N2C U-Net avg | 31.4677 | 0.9760 |
| Median Filter-avg. input | 28.0214 | 0.9495 | *NRRN* | 31.1060 | 0.9708 |
| Gauss Filter-avg. input | 27.0649 | 0.9195 | NRRN -2 img | 30.4194 | 0.9666 |
| NLM -sngl | 24.0703 | 0.6195 | N2C U-Net sngl | 29.1844 | 0.9571 |
| | | | N2C DenoiseNet -sngl | 28.3436 | 0.9559 |

of NLM the high PSNR is achieved by aggressively flattening the signal even on biological structures of interest. NRRN does a better job in removing noise across the resin while sharpening the biological structures - visible on Fig. 1 and Fig. 6b. Additional information is given in the appendix.

Besides the OHSU data, we synthesized additional noise to the EPFL data set. We fine tuned the OHSU pre-trained models by continuing the training on the small EPFL volume data set. During training, we corrupted the EPFL images with Poisson noise for random peak values ranging in $[1, 50]$ and also added Gaussian noise with random $\sigma$ between 10 and 75. The denoising results are in Table 2. In terms of PSNR, NRRN with its unsupervised training shows very comparable performance to the supervised training - marginal improvement in some of the cases and a slight under-performance in others. A closer examination of Fig. 7 reveals that: U-Net tends to over smooth the details; DenoiseNet adds white speckles for low peak values and high $\sigma$, indicating that the network struggles to extract enough information from the significantly damaged images; while NRRN does not display any of these issues because the Building Units are able to synthesize the information from the adjacent slices and reconstruct and remove the noise without over smoothing or adding artifacts.

## 5.2 Exploring the denoising process and ablation study

Our algorithm allows intermediate noise estimates after each Building Unit. Our exploration of the denoising process is carried out for NRRN with 5 BUs on both data sets: OHSU and EPFL. Each BU SSIM shows that the majority of the denoising happens in the first 1-3 layers where we observe a more significant and gradual improvement (see Fig. 8b). The last two BUs become important for highly corrupted images (Poisson noise with peak=1 and Gaussian noise $\sigma = 75$). On Fig. 8a we looked at the signal across the resin after each BU. We scaled the signal with the signal mean value

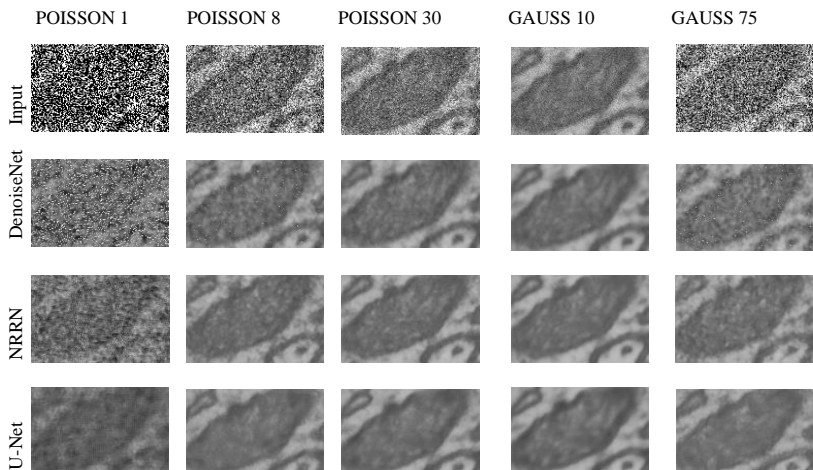

Figure 7: Comparison of the denoising performance. Top to bottom: Input image, DenoiseNet, NRRN and U-Net. Left to right: added Poisson noise with $peak = 1, 8, 30$ and Gaussian noise with $\sigma = 10, 75$.

Table 2: PSNR in dB for EPFL data set with added Poisson noise with peak = 1, 4, 8, 30 and Gaussian noise with $\sigma = 10, 25, 35, 75$

| Method | Poisson noise Peak | | | | Gaussian noise $\sigma$ | | | |
|---|---|---|---|---|---|---|---|---|
| | 1 | 4 | 8 | 30 | 10 | 25 | 35 | 75 |
| Input | 5.98 | 10.19 | 12.52 | 17.66 | 28.13 | 20.17 | 17.29 | 11.54 |
| NLM-avg | 10.16 | 14.72 | 17.03 | 22.05 | 27.98 | 25.78 | 21.55 | 16.11 |
| Gauss-avg | 16.86 | 23.89 | 25.77 | 26.89 | 27.13 | 27.01 | 26.88 | 25.57 |
| Median-avg | 13.54 | 18.26 | 20.14 | 23.71 | 27.17 | 25.09 | 23.53 | 19.11 |
| NRRN | **17.33** | 23.77 | 25.68 | 27.52 | **28.57** | **27.97** | 27.42 | 25.38 |
| N2C DenoiseNet | 15.02 | 22.39 | 25.19 | **27.53** | 28.55 | 27.93 | 27.43 | 24.52 |
| U-net | 16.93 | **23.92** | **25.86** | 27.49 | 27.97 | 27.77 | **27.47** | **25.58** |

and plotted the scaled input image signal (x-axis) versus the scaled BUs signal (y-axis). We see that the signal (which on the resin area is mostly noise) gradually fattens. Looking at the intermediate denoising results one can tune the modular architecture toward the needs in hand: the higher the level of noise, the higher the number of BUs. We trained a whole family of NRRN networks ranging the number of BUs from 1 to 5 on the OHSU data set. Measuring the signal variation on the resin in terms of IQR, PSNR, and SSIM, we observed that while the SSIM and PSNR have not improved, the variation of the signal (in terms of IQR) on the resin flattened until the $4^{th}$ BU and leveled there.

## 5.3 TRANSFER LEARNING AND EFFICIENCY

As we have seen in Table 2 the three architectures achieve similar results on the EPFL data set with pre-training. Our final experiment is skipping the pre-training step and directly training all the architectures on the small EPFL data set. We observed that even with just only 1 BU the NRRN achieves PSNR of $17.9dB$ for the case of Poisson noise with $peak = 1$, while the N2C architectures get only PSNR $= 9.5dB$. Similarly, in the case of Gaussian noise with $\sigma = 10$ NRRN with 1 BU reaches a PSNR of $28.5$ which is at least $4.7\%$ better than the others.

Furthermore, in the Appendix Fig. 11, we report and measure the giga operations per second (GOPS) and number of parameters of the NRRN family (single branch), (N2C) U-Net and (N2C) DenoiseNet. When looking for a trade off between denoising quality and efficiency among the NRRN families,

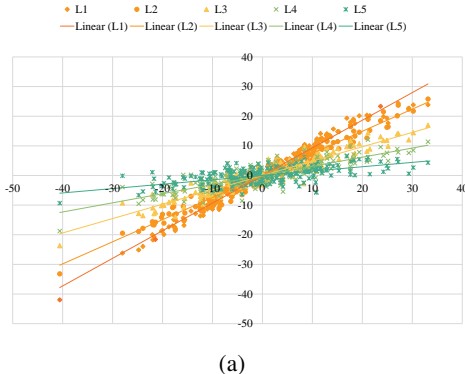
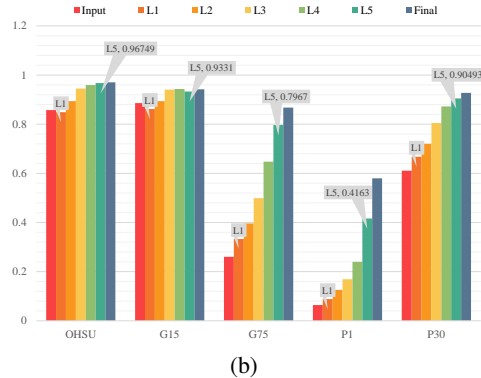

|          (a)          |          (b)          |

Figure 8: Iterative denoising process: a) Input signal scatter plot across the resin vs layers signal, b) SSIM at each layer for OHSU data set and EPFL with added Poisson noise with peak 1 and 30 (P1, P30) and added Gaussian Noise with $\sigma = 15, 75$ (G15,G75)

U-Net and DenoiseNet, the winner is NRRN with 3 Building Units. For more noise corrupted images (as it was for EPFL with Poisson noise with peak 1-8 or Gaussian Noise with $\sigma = 75$) the 5 BUs with their bigger receptive field would be required.

## 6    CONCLUSION

The advances in FIB-SEM provide high resolution cell images, unfortunately encompassing a high level of noise. In this paper, we propose and further developed the idea of using a sequence of images with a Noise2Noise approach for removing that noise. That has the benefit of not requiring ground truth images for training, which is typically the case for biomedical deep learning tackled problems. We suggested a novel modular architecture (NRRN) that is able to exploit the sequential nature of the images and reconstruct and remove the noise from the original image. We demonstrated that this unsupervised approach leads to consistent noise removal across the entire image without regard to the structures present. Besides, we showed that the architecture allows a glimpse into the denoising process that can be used to adjust the depth of the network to the available computational resources and/or presence of noise.

### ACKNOWLEDGMENTS

FIB-SEM data was generated at the Multiscale Microscopy Core with technical support from the OHSU Center for Spatial Systems Biomedicine and invaluable specimen acquisition support from SMMART clinical coordination team.The manuscript was supported by Prospect Creek Foundation, the Brenden-Colson Center for Pancreatic Care, the NCI Cancer Systems Biology Measuring, Modeling, and Controlling Heterogeneity Center Grant (5U54CA209988), the NCI Human Tumor Atlas Network (HTAN) Omic and Multidimensional Spatial (OMS) Atlas Center Grant (5U2CCA233280), the OHSU KCI and NCI Cancer Center Support Grant (P30CA069533), and the OCSSB. This study was approved by OHSU IRB#16113.

### ETHICS STATEMENT

Here with we would like to disclaim a potential conflicts of interest: one of the co-authors has licensed technologies to Abbott Diagnostics; has ownership positions in Convergent Genomics, Health Technology Innovations, Zorro Bio and PDX Pharmaceuticals; serves as a paid consultant to New Leaf Ventures; has received research support from Thermo Fisher Scientific (formerly FEI), Zeiss, Miltenyi Biotech, Quantitative Imaging, Health Technology Innovations and Micron Technologies; and owns stock in Abbott Diagnostics, AbbVie, Alphabet, Amazon, Amgen, Apple, General Electric, Gilead, Intel, Microsoft, Nvidia, and Zimmer Biomet.

REPRODUCIBILITY STATEMENT

We have undertaken the following steps to ensure the reproducibility of our results: the source code is attached as supplemental material, the description of the model, including figures, are in section 3.1, hyper-parameters are in section 5, additional formulas and information are in the Appendix A.1 and A.2.

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

## A    APPENDIX

### A.1    BUILDING UNIT EQUATIONS

For every pair of images $(x_{j\pm1}, x_j)$, the key equations are given by:

$$
\begin{aligned}
\tilde{h}_{j\pm1}^{l+1} &= \tanh(W_1 * \tilde{h}_{j\pm1}^l) \\
u^l &= \sigma(W_u * \hat{h}_j^l + V_u * \tilde{h}_{j\pm1}^{l+1}) \\
r^l &= \sigma(W_r * \hat{h}_j^l + V_r * \tilde{h}_{j\pm1}^{l+1}) \\
c^l &= \tanh(W_c * \hat{h}_j^l + r^l \circ \tilde{h}_{j\pm1}^{l+1} * V_c) \\
\hat{h}_j^{l+1} &= (1 - u^l) \circ \tilde{h}_{j\pm1}^{l+1} + u^l \circ c^l
\end{aligned}
\tag{2}
$$

for $l \in \{0, ..., L\}$, $j \in \{1, ..., N-2\}$ and $\tilde{h}_{j\pm1}^0 = ReLU(W_0 * x_{j\pm1})$ and $\hat{h}_j^0 = ReLU(W_0 * x_j)$ and $W_1$ is with shared parameters across all BUs. Here $\circ$ denotes the Hadamard product and $*$ denotes the convolution operator.

### A.2    LOSS FUNCTION

Wu et al. (2019) showed that for a clean image $s_k$ and two noisy realizations of it $x_{(k,1)} = s_k + n_{(k,1)}$ and $x_{(k,2)} = s_k + n_{(k,2)}$ the function

$$
g(x_{(k,1)}, x_{(k,2)}, \theta) = \frac{f(x_{(k,1)}, \theta) + f(x_{(k,2)}, \theta)}{2} \xrightarrow[\theta \to \theta^*]{} s_k
$$

approaches the real signal $s_k$ for, $\theta^* = arg\min L_{n2n}(\theta)$ where

$$
\begin{aligned}
L_{n2n} = \frac{1}{N} \sum_{k=1}^{N} \{ &\frac{1}{2} ||f(x_{(k,1)}, \theta) - x_{(k,2)}||_2^2 \\
&+ \frac{1}{2} ||f(x_{(k,2)}, \theta) - x_{(k,1)}||_2^2 \\
&- \frac{1}{4} ||f(x_{(k,1)}, \theta) - f(x_{(k,2)}, \theta)||_2^2 \},
\end{aligned}
\tag{3}
$$

The only requirements for such loss function are that the noise $n_{(k,1)}$ and $n_{(k,2)}$ are with zero means and spatially independent.

In our case, we consider $N-2$ triplets of images split into two pairs $(x_{j-1}, x_j)$ and $(x_{j+1}, x_j)$. The NRRN network maps each of these pairs to two outputs $x_j' = f(x_{j-1}, x_j, \theta)$ and $x_j'' = f(x_{j+1}, x_j, \theta)$. The two input pairs with independent and zero mean noise, generate two denoised versions of the same $j^{th}$ image. The final denoised solution $\hat{x}_j$ of the $j^{th}$ image is given by:

$$
\hat{x}_j = \frac{1}{2}(x_j' + x_j'').
$$

One can view, for every slice $x_j$, the adjacent images $x_{j-1}$ and $x_{j+1}$ as discrete versions of the signal along the third direction of the volume. In addition, $x_{j-1}$ and $x_{j+1}$ have their own noise that is spatially independent. Due to the FIB-SEM way of imaging (extremely high isotropic resolution along all three dimensions), we consider that the signal between two slices has good enough properties that we can use a Taylor expansion along the third-axis. Thus, for every image $x_j$ one can view $x_{j-1} = s_j + noise + e_0$, where $s_j$ is the real signal, $e_0 = O(s_{j-1} - s_j)$, and similarly for $x_{j+1}$. In such cases the eq. 3 can be rewritten as

$$
L_{n2n} = \frac{1}{N-2} \sum_{j=1}^{N-2} \left\{ \frac{1}{2} ||x_j' - x_{j+1}||_2^2 + \frac{1}{2} ||x_j'' - x_{j-1}||_2^2 - \frac{1}{4} ||x_j' - x_j''||_2^2 \right\}
\tag{4}
$$

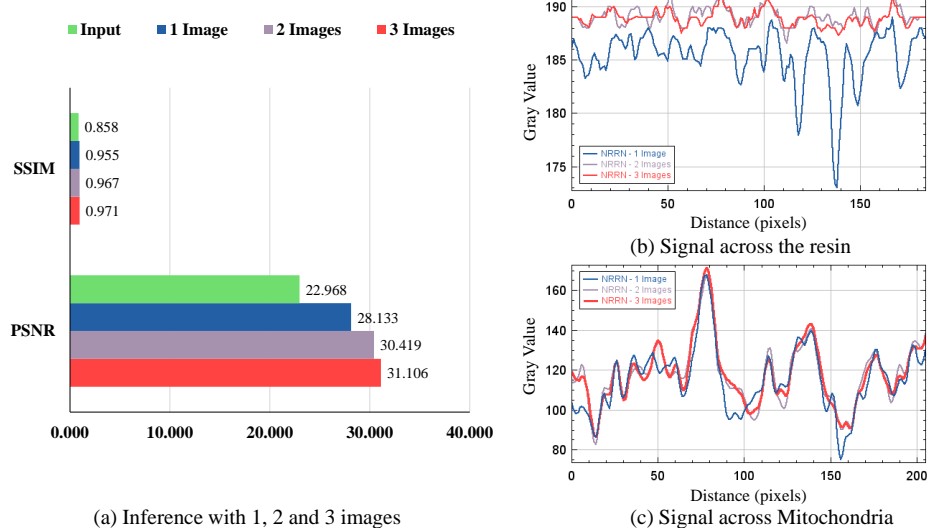

(a) Inference with 1, 2 and 3 images

(b) Signal across the resin

(c) Signal across Mitochondria

Figure 9: Inference with 1, 2, and 3 input images. a) PSNR and SSIM, b) The signal across the resin for inference with one, two and three images, c) The signal across an organelle.

## A.3 RESULTS

### A.3.1 INFERENCE WITH 1, 2 AND 3 IMAGES

NRRN requires a triplet of images in the denoising process, but one can use NRRN for denoising with double or even single inputs during inference. However, as shown in Figure 9, the higher the number of input images, the better the PSNR and SSIM. Even so, an inference with a double input architecture gives very satisfactory quality results in comparison to the triple input by degrading the overall results (measured in PSNR and SSIM) by only 2%. The signal variation in the flat resin area or across the organelle for two images is almost indistinguishable from the three images. Using a single image or two images would, in addition, reduce the computational burden since we would pass the input through the single branch of the architecture.

### A.3.2 EDGE PRESERVING

A good denoising algorithm should be able to suppress noise and at the same time keep the edges sharp. NRRN's architecture removes the noise and does a better job at preserving the edges of the organelles as compared to Non-local mean, U-Net and DenoiseNet (the last two were trained using supervised techniques). In Figure 10 we visually inspect the denoised images (the right panels), the image being passed through high-pass filter (left panel) and the signal variation across a straight line (the bottom graphs).

Additionally, we have experimented with a small modification in our architecture (Fig. 2), exchanging the very first convolution layer for a concatenation of four edge enhancement modules (Liang et al. (2020)) and a convolution with 32 channels. The edge enhancement module contains four trainable Sobel convolutions (detecting vertical, horizontal, and the two diagonal directions edge information). That resulted in a slightly boosted PSNR of 31.12 dB and SSIM of 0.9708 (around 0.3% better than the current architecture). A visual inspection showed no significant enhancement on the edges. Due to the incremental improvement we did not proceed further with that architecture change.

### A.3.3 IQR VS OPERATIONS/PARAMETERS

In Fig. 11, we report and measure the giga operations per second (GOPS) and number of parameters of the NRRN family (single branch), (N2C) U-Net and (N2C) DenoiseNet. If we are to find the best trade off denoising quality versus efficiency, then the NRRN with 3 Building Units gives, in

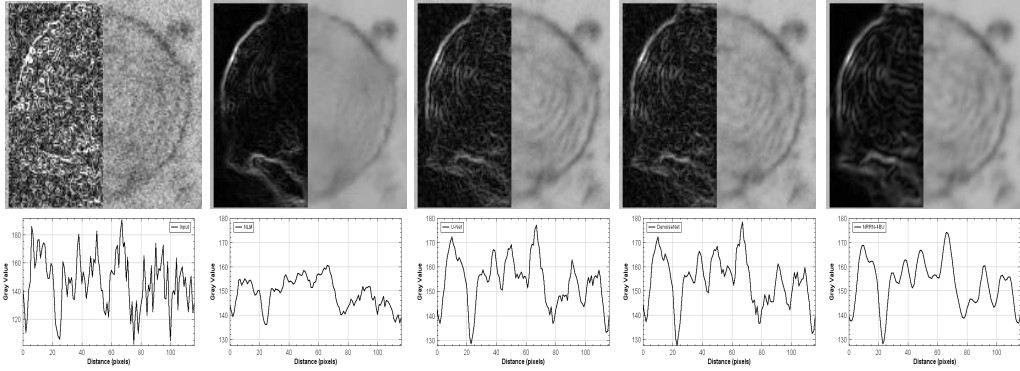

Figure 10: Comparison of the denoising performance (from left to right): Input image, NLM, U-Net, DenoiseNet and NRRN

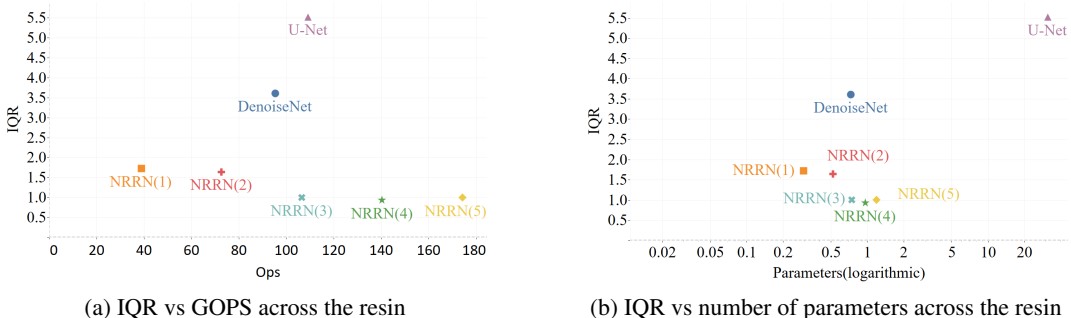

(a) IQR vs GOPS across the resin

(b) IQR vs number of parameters across the resin

Figure 11: Comparison of (a) IQR vs GOPS (b) IQR vs number of parameters (in million) for NRRN family, DenoieNet, and U-Net. For the NRRN family the GOPS are for only one (of the two) branches, the Noise2Noise training requires two branches which will double the GOPS but avoids the need of ground truth images.

our opinion, the best trade off from the whole NRRN family, DenoiseNet, and U-Net for the OHSU data set. For more noise corrupted images (as it was for EPFL with Poisson noise with peak 1-8 or Gaussian Noise with $\sigma$ 75) the 5 BUs with their bigger receptive field would be required.

