# OpenReview forum: "Noise Reconstruction and Removal Network: A New Way to Denoise FIB-SEM Images"
_ICLR.cc/2022/Conference — ICLR 2022 Submitted_

### Official Review · Reviewer_xdev · 2021-10-30

**Correctness:** 3
**Technical Novelty And Significance:** 1
**Empirical Novelty And Significance:** 1
**Recommendation:** 3
**Confidence:** 5

**Main Review:**

-----------------Strengths--------------------------
The paper is specifically aimed at FIB-SEM denoising which is an important application area in the biomedical domain. Although image denoising is a well-studied problem, the impact of image denoising for biomedical images has only been investigated recently. I appreciate the authors looking into this direction.

-----------------Weaknesses---------------------------

1. Novelty: The method proposed in this paper is a variation of Noise2Noise. But I am wondering why would Noise2Noise not work in their case. The authors claim that they need triplets of images (which are consecutive image slices or three images of the same slice) having the same content but different noise realizations. This seems like the perfect case for application of Noise2Noise and that too needing only two images. For instance, Cryo-CARE (Buchholz et al.) uses Noise2Noise in a similar manner fore noising of cryo-electron microscopy images. Why should the same not be applicable to FIB-SEM images?

2. Motivation for architecture and concerns with ablations: As mentioned above, I do not understand the motivation for the proposed architecture. Although the paper claims the novelty of the proposed architecture, I am unsure why is it needed in the first place given we have simpler and theoretically well-founded methods such as Noise2Noise. Additionally, the proposed NRRN has many modules and it is hard to grasp which module is responsible for NRRN's performance. For example, how useful is the Noise Attention block, Feedback block and the Synethesizer block? How do gating blocks affect the performance in the BU module? I think there are no reported ablations for these. Overall, I do not understand the motivation for the complexity of the proposed architecture.

3. Baselines: The proposed method has not been compared to many important baselines. Many of these baselines such as  Self2Self (Quan et. al) , Noise2Same (Xie et al.), Cryo-CARE (Buchholz et al.), PN2V (Krull et al.), PPN2V (Prakash et al.), etc. will be directly applicable to the datasets considered in this paper with no modification. Additionally, these baselines will not need triplet of images as proposed in this paper and most of them can be applied on single noisy images as well.

4. Related Work: Image denoising is a well-studied problem and the paper does not acknowledge many important prior works in general as well as in the biomedical image denoising domain. For instance, the related work section does not mention popular non-DL based methods such as BM3D, NLM, etc (although they have been used as baselines). Additionally, many recent DL based methods such as Self2Self (Quan et. al) , Noise2Same (Xie et al.), Cryo-CARE (Buchholz et al.), PN2V (Krull et al.), PPN2V (Prakash et al.), etc. have also not been mentioned or compared against even though all these methods are unsupervised denoisers targeting same noise types (Gaussian and Poisson) as studied in this paper.

5. Limited Applicability: As the paper admits, the proposed method is specifically designed for FIB-SEM images which although being important is a limited class of biomedical image domain. There are many works (see my comments on ```Related Work```) which target the same noise types but are applicable to different biomedical image domains as well as natural images.

**Summary Of The Paper:**

The paper proposes a denoising algorithm for FIB-SEM images using triplets of images. These triplets can either be consecutive imaged slices or three scans of the same slice. The proposed NRRN network takes two sets of image pairs (constructed from triplets) and generates two denoised images corresponding to the two sets with the final denoised solution being the average of the two denoised images. The method is tested on two datasets and compares against few existing supervised denoising methods.

**Summary Of The Review:**

In summary, I vote for rejecting this paper.

The paper misses comparisons against many important and recent unsupervised denoising works which are directly applicable for the presented application domain without needing any modifications (see my comment in weaknesses). A motivation for choosing the proposed architecture is missing (see my comment in weaknesses) when in my opinion, simpler architectures can arguably perform similar or better. Moreover, the paper is rather limited in its application (is specifically designed for FIB-SEM images) and is unclear if it can be extended for any other domains. Hence, my rating of reject.

---

### Official Review · Reviewer_34aJ · 2021-11-01

**Correctness:** 3
**Technical Novelty And Significance:** 2
**Empirical Novelty And Significance:** 2
**Recommendation:** 5
**Confidence:** 3

**Main Review:**

The strengths of the current work include addressing the challenging biomedical application of denoising *FIB-SEM* images. A novel recurrent neural network architecture employing soft attention is proposed and the *N2N*- loss formulation is modified in order to suit the current biomedical application.

I have gone through sections 1, 2, 3.2, 4, 5 and Appendix A.2 and in my opinion, the weaknesses of the paper (without being able to look under the hood at any code since it was not made available by the authors as of now) is as follows:
1. It would be interesting to see how well this loss formulation performs with a U-Net like network architecture just in order to distinguish the contribution of the loss formulation and the architecture towards the performance of the method. Additionally, it would be interesting to investigate  the same  recurrent neural architecture employed in this work but training the model parameters with a N2V (*Noise2Void, Krull et al*) blind spot  strategy, for example.

2. In A.2, the authors state that Taylor’s expansion is used
>“Due to the FIB-SEM way of imaging (extremely high isotropic resolution along all three dimensions), we consider that the signal between two slices has good enough properties that we can use a Taylor expansion along the third-axis.”

* Here, the third-axis should be clarified to be the axis of milling.
* Furthermore, the authors should clarify that they use a zeroth-order Taylor Approximation and ignore all higher order terms.
* This is a more strong weakness in my opinion - *NRRN* rests on the assumption that the signal in any slice is the same as the signal in the slice before and the signal in the slice after, since the authors ignore all higher order terms in the Taylor’s expansion. Hence, the authors should highlight that this method starts to break down if the voxel size along the  z or third dimension is large. Probably an ablation study highlighting this drawback by considering every second slice in the *EPFL* dataset to form a triplet would highlight this.

3. It would be useful to compare against baseline methods designed especially for electron microscopy image data such as [Buchholz et al](https://pubmed.ncbi.nlm.nih.gov/31326025/) , if possible.

4. Rather few slices during evaluation in the OHSU dataset- the OHSU dataset comprises 5 slices, 3 of which are used for training, 1 for validation and 1 for testing. Since their proposed approach relies on feeding in a triplet of images to the network, hence feeding in only one test image during evaluation seems somewhat ill-suited to show the merits of the proposed method.

5. Minor point: The EPFL dataset is commonly referred to as the `Lucchi` dataset in some publications (for example, [this](https://arxiv.org/pdf/2104.03577.pdf) or [this](https://ieeexplore.ieee.org/document/6619103) ) after the surname of the first author. The correct reference for this dataset should be *Lucchi, A., Smith, K., Achanta, R., Knott, G., Fua, P.: Supervoxel-based segmentation of mitochondria in em image stacks with learned shape features. IEEE Transactions on Medical Imaging 31(2), 474–486 (2011)*

6. Minor point, regarding  the **Introduction** section:
>“Theoretically, obtaining denoised images is possible by averaging multiple (up to hundreds) acquisitions of the same sample. As mentioned, this is not feasible with FIB-SEM.”

I think that the reason why several acquisitions at the same depth are not acquired in the *FIBSEM* context can be better motivated in the introduction section perhaps by stating explicitly that it leads to sample degradation/damage, the idea of a finite electron budget can be introduced perhaps etc.

7. Minor point, regarding the **Results and Comparison** section:
>“Initial large images were cropped into 13650 smaller images”

How is the cropping done? Would be nice to mention that the crops were generated randomly with or without any overlap, etc

8. Minor point, regarding the **Dataset and Metrics** section:
>“The process was repeated 5 times”.

This statement was not clear enough to me- does this mean that the ground truth was generated from 50 images of the same area? An explanation to this effect would be welcome.

9. Minor point, regarding the **Denoising Module** subsection:
>“L+1 stacked layers made of a building unit (BU) coupled with noise attention (Natt) and feedback blocks”.

What does the variable `L` stand for? It would be good  to introduce it before in the text, prior to using it in the aforementioned sentence.

10. Minor point: The authors state that the “the source code is attached as supplemental material”, but that doesn’t seem to be the case yet.

11. Minor point: Some of the figures (Fig. 6, 8, 9, 10) are hard to read: small font and smudgy bitmaps.







**Summary Of The Paper:**

The authors propose an unsupervised learning framework called `NRRN` for denoising focused ion beam scanning electron microscopy  (*FIBSEM*) images - which are typically acquired with a high resolution along all dimensions - without requirement of any clean ground truth data during training. They do so by considering a triplet of images obtained from neighboring consecutive slices, employing a mean-squared error like loss formulation inspired from *Lehtinen et al, 2018 (N2N)* and *Wu et al, 2019* and proposing a novel recurrent neural network architecture. The authors show comparable results to supervised denoising approaches (which have access to clean ground truth data during training) on the *OHSU* dataset and on the simulated-noisy dataset from *EPFL*.

**Summary Of The Review:**

`NRRN` is a novel approach to address the challenging application of denoising *FIB-SEM* images. The authors back this up with state of the art results on a real and a noisy-simulated dataset. I rate this work slightly below acceptance threshold currently because even though the solution to this difficult biomedical problem  is very welcome, it is not evident to me how much of the performance of the method owes itself to a new architecture and how much benefit is gained by the employed loss formulation.  Furthermore, because of the zeroth-order approximation to the Taylor series along the z (third) axis implies to me that as soon as the voxel size along this dimension increases, the results would become sub-par. While this is okay and not a problem perhaps in the *FIB-SEM* context (?) I think it would be important to mention this explicitly or comment on how this can be fixed/handled if one were to extend this method and the proposed architecture to other domains. The writing in general is good but at times lacks some motivating sentences leading to the argument. Since the work concerns this specific biomedical application, perhaps the authors can consider submitting their work to a more domain-relevant conference or journal, in order to accrue a higher impact and relevant target audience.

---

### Official Review · Reviewer_aPxW · 2021-11-02

**Correctness:** 3
**Technical Novelty And Significance:** 2
**Empirical Novelty And Significance:** 2
**Recommendation:** 5
**Confidence:** 4

**Main Review:**

Strengths: biomedical imaging is one of the most important application areas of computer vision and machine learning. Designing methods that take into account the specific properties of a certain type of biomedical imaging is surely impactful and thus welcomed. The authors clearly demonstrate their expertise in FIB-SEM imaging. This method explores unsupervised models on very large high-resolution images, which is challenging.

Weaknesses: Although this biomedical application is meaningful, the novelty behind this method is limited especially for a major ML conference. Other than the loss function, which is only a minor novel design, the proposed method is mainly an adaptation of an established approach. The authors did the experiments on only two datasets. I am not sure how generalizable is the proposed method to other FIB-SEM datasets or other similar imaging domains.

Most importantly, the experiments showing structural details are preserved whereas noises are removed, are missing. The two metrics, PSNR and SSIM are essentially distance functions which are dominated by low-frequency features. However, the main advantage of learning-based methods over traditional methods, such as the Gaussian filter, should be to distinguish high-frequency features from noises. Since the ground truth is available for one dataset, I would suggest the authors to only keep the high-frequency region of the ground truth and compare with different denoising results.

Minor concerns:
1. Texts too small in Figure 1-4, 6, and 8.
2. Related works should be better organized by including subsections.
3. It is unclear how section 5.3 is related to transfer learning.


**Summary Of The Paper:**

This paper proposes an unsupervised denoising convolutional neural network model for Focused Ion Beam-Scanning Electron Microscopy (FIB-SEM) images. The whole framework is adapted from the Noise2Noise approach. The proposed architecture Noise Reconstruction and Removal Network (NRRN) contains stacked layers of building units and noise attention modules to gradually denoise the input image. Additionally, a loss function, modified from the original loss function of Noise2Noise is specifically designed for FIB-SEM image taking into account its imaging properties. The results are demonstrated on two datasets, one public and one private, which outperformed both traditional and learning-based baselines. An ablation study is performed by varying the noise type and noise level.

**Summary Of The Review:**

It is a good work overall with a clear presentation. However, the novelty is limited and more experiments should be performed to show the utility and advantage of the proposed method.

---

> ### Author Response · Authors · 2021-11-23
> **Answers to the raised questions**
>
> Thank you for taking the time and for the positive feedbacks.
> 1. The main contribution for us is the building unit to reconstruct and remove the noise. In our ablation study and Figure 8 we show how having a modular architecture with our building units can be stacked up or slimmed down based on the need for a given dataset/requirement.
> 2. We agree that we should also look into high frequency features. Hence, we also provide empirical results in Figure 1, 10, and supplementary video.

---

> > ### Comment · Reviewer_aPxW · 2021-11-29
> > **Response to the authors**
> >
> > Thanks for the answer. Some of my concerns have been addressed.

---

### Official Review · Reviewer_Uq3h · 2021-11-02

**Correctness:** 3
**Technical Novelty And Significance:** 2
**Empirical Novelty And Significance:** 2
**Recommendation:** 5
**Confidence:** 4

**Main Review:**

The text does a good job of introducing the problem and providing a short review of prior work in this area. The paper is overall well written and easy to read.

The authors propose to use triplets of consecutive sections in a noise2noise setting with a modified loss, and a custom architecture composed of stacked Build Units and Noise Attention Blocks, reducing noise at every level in the stack. The idea of using multiple consecutive sections in an image stack has been proposed before (https://arxiv.org/abs/2011.05105), and this should be mentioned (and ideally compared to) in the present work. Evidence that the BUs really do reduce noise with subsequent application is shown in Fig. 8, though it would be helpful to also see this visually (perhaps in the supplementary materials if space is a concern). It would also be interesting to see a visualization of the noise that is removed at every step.

The denoising results are compared to training-free methods, as well as supervised baselines. The proposed model performs comparably to the supervised methods in terms of PSNR and SSIM, and where the other models perform better, the authors argue that this is because they learn to reproduce the noise still present in the ground truth data. The improvements offered by the proposed NRRN model are somewhat subtle -- e.g. fewer artifacts and flatter profiles in empty (resin) areas. This is shown in Fig. 6. I found the attached movie very helpful to intuitively understand the impact of this, so it could be helpful to include some high-res figures in the text as well. I also appreciated the computational complexity comparison to DenoiseNet.

The paper would be strengthened by including a noise2noise baseline (this should address the concern of the baseline models learning to reproduce ground truth noise). It would also be helpful to have some ablations of the proposed BU architecture. The architecture is quite complex, and there is no experimental data showing that e.g. the LSTM-inspired internal gates are actually necessary.


Questions:
- Has it been tried to modify the building unit to take 2 sections of context directly instead of feeding a section pair at a time and averaging the results?
- What is the total number of images and Mpx contained in the dataset?
- Please comment on how section alignment quality impacts the denoising results.
- Please comment on image degradation due to electron irradiation in case of multiple acquisitions. Is this a problem at all within the 10 scans you did?
- A.3.1 mentions inference with a single image. How does this work, given that 2 inputs are shows in Fig. 2?

Minor nits:
- Fig. 5 does not show the yellow dotted line mentioned in the caption.
- The paper mentions that supplementary materials contain source code, but only a single AVI
file is present.



**Summary Of The Paper:**

The paper proposes a neural network architecture (NRRN) implementing a variant of the noise2noise scheme, and applies this to section triplets of a FIB-SEM stack. The results are evaluated with PSNR, SSIM, and IQR on a custom dataset with low-noise ground truth, as well as on a public dataset with no ground truth (using artificially introduced noise). NRRN is shown to improve the images in ways that are hard to measure with traditional metrics.


**Summary Of The Review:**

The proposed neural network architecture and training scheme improve denoising over baselines in relatively subtle ways that are hard to measure with traditional metrics. To me, in the current version it slightly misses the bar for acceptance due to some missing baselines (noise2noise, noise2stack) and insufficient experimental data motivating the architecture choices.

---

> ### Author Response · Authors · 2021-11-18
> **Answers to the posed questions**
>
>
> *   *Q: Has it been tried to modify the building unit to take 2 sections of context directly instead of feeding a section pair at a time and averaging the results?*
> A: No, we have not tried that.
> *  *Q: What is the total number of images and Mpx contained in the dataset?*
> A: In S5: after patching the training  data set consist of 812 images  of size  256x256 Mpx .
> *  *Q: Please comment on how section alignment quality impacts the denoising results.*
> A: We do expect that section alignment will impact the quality, but no further experiments have been made to quantified the  impacts.
> *  *Q: Please comment on image degradation due to electron irradiation in case of multiple acquisitions. Is this a problem at all within the 10 scans you did?*
> A: The Images do degrade in the case of multiple acquisition. Collecting 10 scans per surface requires a highly trained technician,  obtaining  3 scans is a much easier task.
> *  *Q: A.3.1 mentions inference with a single image. How does this work, given that 2 inputs are shows in Fig. 2?*
> A: Inference with a single image works with making forward pass with two copies of the same image. The results are shown in the A.3.1. The unsupervised NRRN with single image gives PSNR of 28.1 which could be compared to the supervised training DenosieNet 28.3, and supervised U-Net 29.2. That is to say that the PSNR is slightly worst or comparable to the supervised training.
> *  *Q: Fig. 5 does not show the yellow dotted line mentioned in the caption.*
> A: Our mistake, the dotted line is in gray color.
> *  *Q: The paper mentions that supplementary materials contain source code, but only a single AVI file is present.*
> A: https://github.com/codeAC29/NRRN

---

> > ### Comment · Reviewer_Uq3h · 2021-11-21
> > **Thanks**
> >
> > Thank you for taking the time to answer the questions from the review!

---

### Official Review · Reviewer_4BoC · 2021-11-04

**Correctness:** 2
**Technical Novelty And Significance:** 2
**Empirical Novelty And Significance:** 2
**Recommendation:** 3
**Confidence:** 3

**Details Of Ethics Concerns:**

.

**Main Review:**

Pros:
+ Noise in the FIBSEM, and more generally in the EM images, is a real problem and many approaches in different applications would benefit from a noise removal technique.
+ An unsupervised technique for denoising is also highly desirable.
+ I must also appreciate the effort to design an architecture and loss functions for this task.

Cons:
- **Generality:**  However,  I am concerned the utility of this method on different types of medical images. In fact, the assumption of structural continuity in the z dimension or having multiple images taken from the same region of tissue may be too restrictive for the EM images itself. The SEM and TEM imaging approaches, that are widely popular because their capability to record larger tissue volume, has a z-resolution of approx ~30nm. The contents of two slices in these types of images vary significantly and violates the working assumption of this method.

As FIBSEM imaging destroys the tissue samples, recording multiple images from the same region may not possible in practice, not only because of scarcity of the same tissue but also because the extensive, elaborate sample preparation necessary to produce images of the same quality.

Not sure if the requirement of having multiple slices of the same region or consecutive slices with very similar content can be satisfied in other medical images or not.

- **Experimental validation:** Since there exist other methods (Lethinen'2018, Wu'2019) already in the literature that removes noise in an unsupervised fashion, the proposed method must be compared with both or at least one of these algorithms. I could find this comparison in the manuscript.

- **Technical/conceptual/theoretical justification:** Although the paper describes the network architecture in Sec 3.1, it does not explain why and how the architecture is removing noise. Why and how do "we obtain a less noisy image x'_j^l after each layer" conceptually/theoretically or ate least intuitively? What is the conceptual/theoretical reason "the output of the NAtt block represents component of the noise present in previous noisy image x'_j^{l-1}"? How does the synthesizer learn the noise components? Why is the attention map multiplied by the *first* layer of \hat{h}_j_{l+1}? Without proposer justification, it sounds like a trial-and-error strategy for the network design.

- **Presentation:** The description of Sec 3.1 was not easy for me to understand. It would be better to have the equations shown here rather than the appendix. That way, perhaps all the components would be accounted for -- now the description misses c^l, c^{l+1}, it does not what they are and how are they computed. There is also some inconsistency in the statements and figures. For example, Fig 4(a) shows connection between synthesizer and NAtt, but the text mentions the upper block is the synthesizer which should not have any connection with NAtt according to Fig 3.

**Summary Of The Paper:**

This paper proposes a method for noise removal from the FIBSEM images. The method is unsupervised, it does not require a corresponding clean image to learn denoising. Instead the method utilizes consecutive slice from a FIBSEM volume or multiple images from the same region to remove noise. The core assumption is, while the noise pattern is random and varies across different z-slices, the actual content, i.e., the cellular structure it is capturing remains largely similar.

The paper proposes a network architecture and loss functions for this purpose. The architecture is is inspired by the LSTM/GRU designs, perhaps in the hope of modeling the common structure between a pair of images.

The approach has been tested on two real datasets.

**Summary Of The Review:**

My current understanding is, the paper lacks the general applicability and sufficient experimental validation that are essential for acceptance in ICLR.

---

### Decision · Program_Chairs · 2022-01-20

**Decision:**

Reject

**Comment:**

This paper addresses the challenging application of denoising FIB-SEM images. State-of-the-art results are reported on a real and a noisy-simulated dataset. Unfortunately, this paper failed to convince the reviewers and received 4 negative ratings. The paper misses critical comparisons against baselines and appears rather limited in scope. The authors failed to provide adequate answers to some of the reviewers' points.